# Toxicity Detection in Finnish Using Machine Translation

**Anni Eskelinen, Laura Silvala, Filip Ginter, Sampo Pyysalo and Veronika Laippala**
TurkuNLP
University of Turku, Finland
`aeeske, laura.s.silvola, figint, sampo.pyysalo, mavela`
`@utu.fi`

## Abstract

Due to the popularity of social media platforms and the sheer amount of user-generated content online, the automatic detection of toxic language has become crucial in the creation of a friendly and safe digital space. Previous work has been mostly focusing on English leaving many lower-resource languages behind. In this paper, we present novel resources for toxicity detection in Finnish by introducing two new datasets, a machine translated toxicity dataset for Finnish based on the widely used English Jigsaw dataset and a smaller test set of Suomi24 discussion forum comments originally written in Finnish and manually annotated following the definitions of the labels that were used to annotate the Jigsaw dataset. We show that machine translating the training data to Finnish provides better toxicity detection results than using the original English training data and zero-shot cross-lingual transfer with XLM-R, even with our newly annotated dataset from Suomi24.

## 1 Introduction

Social media is filled with moderated and unmoderated content with foul language such as threats, insults and swears. Due to the popularity of the platforms and the sheer amount of comments, posts and other user-generated content they include, moderation by human-raters is getting impossible. This makes automatic toxicity detection a requirement in the monitoring of social media platforms and other online settings in order to guarantee a safe and friendly digital space.

In recent years, many studies have tackled the detection of toxic language as well as other similar and relevant tasks, such as the detection of hate speech and offensive language (Davidson et al., 2017; MacAvaney et al., 2019). However, most datasets and thus most of the studies focus on English, leaving other languages with very scarce resources (Davidson et al., 2017; Androcec, 2020). At the same time, the development of the resources, in particular the creation of manually annotated training data, is very time-consuming. Cross-lingual transfer learning has offered a solution to this challenge by allowing the use of data in one language to predict examples in another one. This method has showed promising results in tasks such as register labeling (Rönnqvist et al., 2021; Repo et al., 2021) and offensive language detection (Pelicon et al., 2021). Additionally, recent advances in machine translation open up the question of how to use machine translation to do the language transfer and create novel resources for a language.

In this paper, we address the lack of resources for toxicity detection in languages other than English by benefitting from the recent advances in machine translation. Specifically, we present the first publicly available dataset for toxicity detection in Finnish that we develop by machine translating the English Jigsaw Toxicity Dataset that is claimed to be the biggest and most widely used toxicity dataset (Androcec, 2020). We show that machine translating the dataset to Finnish provides better results for toxicity detection than cross-lingual transfer learning, where a cross-lingual XLM-R model (Conneau et al., 2020) is fine-tuned using the original English Jigsaw training set and tested on the Finnish machine translated test set. Furthermore, to test how much machine translation modifies the content of the dataset and thus causes performance loss, we backtranslate the dataset from Finnish to English, demonstrating only a minimal decrease in performance. Finally, in order to examine how much toxic content the trained model identifies from another source

than the Wikipedia edit comments included in Jigsaw, we create another test set for toxicity detection in Finnish by manually annotating comments from the Finnish discussion forum Suomi24 and building a dataset of 2,260 comments. The annotations follow the label description guidelines that were used to annotate the original English dataset. We show that while the model does identify toxic content also from the discussion forum comments, the change of text source does present some challenges.

As machine translation systems, we test two systems to see whether there are major differences to our results: the DeepL machine translation service[1] and Opus-MT (Tiedemann and Thottingal, 2020), see Section 3.2. The DeepL machine translated dataset, the native Finnish dataset and the resulting fine-tuned FinBERT large model are all openly available at the TurkuNLP Huggingface page[2].

## 2   Related Work

**Toxicity**, in terms of speech, text or behaviour, is an umbrella term that encompasses many kinds of language use, such as hate speech, abusive language, offensive language and harmful language. In this paper, we follow the definition adopted in the Jigsaw dataset and define toxicity as "rude, disrespectful, or unreasonable language that is likely to make someone leave a discussion" (Jigsaw).

**Toxicity detection** (van Aken et al., 2018; Pavlopoulos et al., 2020; Burtenshaw and Kestemont, 2021) is related to many other similar classification tasks such as hate speech detection (Davidson et al., 2017; MacAvaney et al., 2019) and offensive language classification (Davidson et al., 2017; Jahan and Oussalah, 2020). In all these tasks, the goal is to identify harmful text in, e.g., social media, where comments can be flagged for review or automatically deleted.

**Toxicity datasets** and datasets for other related tasks are mostly monolingual with English being the most popular—most studies have used the same Jigsaw dataset that we use (Androcec, 2020). For instance, Carta. et al. (2019) reported ROC_AUC-scores of nearly 90% on this dataset. Additionally, datasets are available, e.g., for Spanish (Androcec, 2020), and a multilingual dataset has been developed as a part of the Kaggle competition on multilingual toxicity detection[3].

The available datasets represent various domains and text lengths, ranging from short Twitter posts (Davidson et al., 2017) to Wikipedia editor comments featured by the Jigsaw dataset we are using, see Section 3.1. Similarly, the annotation strategies vary from multi-label annotation where one instance can have several independently assigned labels to multi-class where one instance can be assigned just one label (Davidson et al., 2017) and to even a binary setting where each instance is either clean or toxic (D'Sa et al., 2020). Due to these differences, combining several datasets to increase the number of examples in training data is difficult.

Similarly, the subjectivity entailed in toxicity creates a challenge for its automatic detection—as people interpret things differently, a single correct interpretation of a message as toxic or not may not exist (see discussion in Ross et al. (2016)). In addition to model performance, the subjectivity can be noted in low inter-annotator agreements. For instance, Waseem (2016) reported a kappa of .57, which can be interpreted as *weak*.

**Cross-lingual zero-shot transfer learning** where the model is trained on one language and tested on another relies on multilingual language models that have been trained on massive amounts of multilingual data (Conneau et al., 2020; Devlin et al., 2018). These have been used for the zero-shot cross-lingual transfer of hate speech detection and offensive/abusive language detection. For instance, Pelicon et al. (2021) report that a multilingual BERT-based classifier achieves results that are comparable to monolingual classifiers in offensive language detection and also Eronen et al. (2022) demonstrate that zero-shot cross-lingual transfer can achieve competitive results for abusive language detection. However, Nozza (2021) note also challenges—the zero-shot transfer of hate speech detection can be complicated by non-hateful, language-specific taboo interjections that are interpreted by the model as signals of hate speech, and Leite et al. (2020) also found that zero-shot transfer did not produce accurate results for toxicity detection in Brazilian Portuguese.

**Machine translation** can be considered a mode of transfer learning that has become viable with the advances of natural language processing. In

---

[1]https://www.deepl.com/translator
[2]https://huggingface.co/TurkuNLP

[3]https://www.kaggle.com/c/jigsaw-multilingual-toxic-comment-classification

particular, the method has been used in toxic language detection to get more data by data augmentation (Rastogi et al., 2020) and by translating data to English to be able to use ready-made models (Kobellarz and Silva, 2022). Kobellarz and Silva (2022) found that comments that were analyzed as toxic in Portuguese were not as toxic when translated to English—however, the same behaviour may not apply to other language pairs. To our knowledge, no experiments comparing cross-lingual transfer by a multilingual model and by machine translation have been made previously.

## 3 Data and Translation

### 3.1 Jigsaw Toxicity Dataset

The data used in this paper is the Jigsaw dataset developed by Google and released as a Kaggle competition[4]. The dataset is based on comments from Wikipedia's talk page edits and consists of 223,549 comments. The dataset collection was done by crowd-sourcing. No specific information about the annotation process is given.

The annotation scheme is composed of six classes: *toxicity*, *severe toxicity*, *identity attack, insult, obscene* and *threat*. Toxicity is a general label encompassing all toxicity and is defined as "rude, disrespectful, or unreasonable language that is likely to make someone leave a discussion", and severe toxicity as "a very hateful, aggressive, disrespectful comment or otherwise very likely to make a user leave a discussion or give up on sharing their perspective". For other definitions, please see the annotation guidelines for Perspective API (Perspective, a,b).

The annotation is set up as multi-label, where each comment annotated as toxic has one or more labels assigned to it. The label distribution of the dataset is presented in Table 1. In total, only 11% of the comments are annotated with at least one of the toxic labels, the rest being left without labels and considered as neutral or non-toxic. This means that the label distribution is highly unbalanced, which, however, comes from the nature of the data as most comments are neutral in discussions. More information about label co-occurrence is given in Figure 1, showing that in particular *obscene* and *insult* as well as *toxicity, insult* and *obscene* co-occur.

---

[4]https://www.kaggle.com/competitions/jigsaw-toxic-comment-classification-challenge/

|                 | Train   | Test   |
|-----------------|---------|--------|
| Toxicity        | 15,924  | 6,090  |
| Severe toxicity | 1,595   | 367    |
| Threat          | 478     | 211    |
| Obscene         | 8,449   | 3,691  |
| Insult          | 7,877   | 3,427  |
| Identity attack | 1,405   | 712    |
| No label        | 143,346 | 57,735 |

Table 1: Label distribution in the Jigsaw Toxicity Dataset. As each comment may have up to six labels, the total number of labels exceeds the number of comments in the dataset.

The dataset is split into train and test sets with stratified sampling (159,571 and 63,978 comments) following the original Kaggle release. Furthermore, for our training purposes with the Finnish data, a development set is split from the train set by doing stratified splitting and taking 20% of the train set comments.

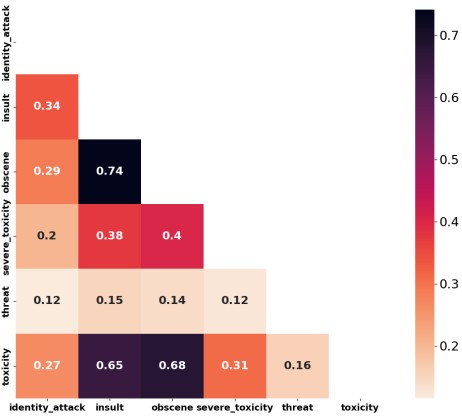

Figure 1: Correlation matrix of the labels of the original train dataset calculated with Pearson standard correlation coefficient. Small values close to zero indicate no correlation between the labels, while higher values closer to 1 suggest correlation and that the labels tend to appear together.

### 3.2 Jigsaw Toxicity Data in Finnish

We machine translated the original English Jigsaw dataset to Finnish using two translation tools: the DeepL machine translation service[5] and Opus-MT (Tiedemann and Thottingal, 2020). For DeepL, the dataset was converted to the required `.docx`-format for the translation and then back to `.jsonl` after the translation. The English-

---

[5]https://www.deepl.com/translator

Finnish translation cost less than 100 dollars. None of the comments were lost during this process—a possibility that needs to be considered when transforming data from a format to another.

For Opus-MT, the texts needed to be sentence split, because the tool can only translate one sentence at a time. This was done using the Udpipe REST api[6]. The model used for translation was `Helsinki-NLP/opus-mt-tc-big-en-fi`[7]. Some of the comments from the test set did not survive the translation, as they were not in English. These were edited to only include the notion "EMPTY".

Finally, to examine the loss of performance caused by the machine translation, we also **back-translated** the dataset translated with DeepL from Finnish back to English. This was done using the same method as the English-Finnish translations.

The DeepL machine translated dataset is available at the TurkuNLP Huggingface[8].

### 3.3 Native Finnish Toxicity Dataset

To examine how much toxic content the fine-tuned model can identify in comments featuring another text variety than the Wikipedia editor comments included in Jigsaw, we developed a new manually annotated test set sampled from Suomi24—the largest online discussion forum in Finland compiled into a giga-size corpus. As the label distribution is very skewed in the Jigsaw dataset with a large majority of comments not annotated for toxicity (see Section 3.1), the sampling was done in a specific manner to ensure a representative set of comments featuring varying degrees of toxicity and the six toxicity classes.

Specifically, we first classified 945,867 comments taken from Suomi24 using a model that was at the time our best performing model which was a fine-tuned base model of FinBERT (Virtanen et al., 2019). Then, for each of the six toxicity labels, we binned the comments to ten bins based on the classifier score for that label (0.0-0.1, 0.1-0.2, ... 0.9-1.0). The distribution of comments in these bins is presented in Appendix A, showing that the classifier is very certain about most of its decisions. In particular, the 0.0-0.1 bins are ex-

tremely large, while another set of peaks can be seen on the right end with high scores.

After the binning, we selected randomly 50 comments from each bin for annotation. This gave 500 comments of broadly varying degrees of predicted toxicity for each of the six toxicity labels. Each of the six batches of 500 comments were annotated for one toxicity label only. Thus, the annotations are multi-class instead of the original Jigsaw multi-label, although 23 individual comments were selected in two different batches due to the sampling for each label being independent. This also means that a comment can have some other type of toxicity that was not annotated for that specific comment.

|                 | Label | No label |
|-----------------|-------|----------|
| Toxicity        | 158   | 193      |
| Severe toxicity | 25    | 328      |
| Threat          | 40    | 391      |
| Obscene         | 170   | 239      |
| Insult          | 145   | 219      |
| Identity attack | 131   | 221      |
| Total           | 669   | 1591     |

Table 2: Label distribution in the native Finnish annotations.

The annotation was done independently by three native Finnish speakers with borderline cases jointly resolved and documented. This process resulted in guidelines which include general directions for the labels meaning the guidelines can be used for any language as a starting point for annotation. For the initial process of annotating a label, we annotated 100-200 comments and used the definitions of the labels found in the Perspective API (Perspective, a,b) as a starting point, after which we had a discussion where we added our own specifications to the guidelines. Then the last 300-400 comments were annotated according to those guidelines.

The inter-annotator agreement for the initial annotation and the annotations done after the discussion can be found in Table 3. As can be seen, the unanimous agreement is very low in almost every label category, which is common for toxicity datasets as mentioned in Section 2. *Threat* is the only label with a higher agreement of around 80% whereas most of the other labels range between 47 and 66%. Unfortunately, our mean agreement did not get better after the discussion which once again shows the difficulty of the task.

The final dataset was formed using only the comments that were initially unanimously labeled

---

[6]https://lindat.mff.cuni.cz/services/udpipe/api-reference.php

[7]https://huggingface.co/Helsinki-NLP/opus-mt-tc-big-en-fi

[8]https://huggingface.co/datasets/TurkuNLP/jigsaw_toxicity_pred_fi

| | Initial | After discussion |
|---|---|---|
| Toxicity | 58% | 54% |
| Severe toxicity | 63% | 66% |
| Threat | 82% | 80.3% |
| Obscene | 69% | 62% |
| Insult | 47.5% | 49.6% |
| Identity attack | 54.5% | 66.6% |
| Mean | 62.3% | 63% |

Table 3: Unanimous inter-annotator agreement (IAA) for the native Finnish toxicity dataset

or for which the label was resolved in a subsequent discussion. While the initial annotations showed significant divergence, this filtering protocol should assure the internal consistency and validity of the dataset. Altogether, the final dataset consists of 2,260 comments natively written in Finnish, further described in Table 2. The guidelines created during the annotation process are published together with the dataset on Huggingface[9].

## 4 Fine-tuning

We use both monolingual and multilingual state-of-the-art models in the detection experiments. Specifically, the monolingual models are the large and cased versions of the original BERT for English (Devlin et al., 2018) and Fin-BERT for Finnish (Virtanen et al., 2019). For the crosslingual experiments, we use XLM-RoBERTA (XLM-R) Large (Conneau et al., 2020) because it has been shown to provide better results than the multilingual BERT for many tasks (Repo et al., 2021; Rönnqvist et al., 2021).

All the experiments are done in a multi-label setting. However, when evaluating classifier performance on the native Finnish test set where the comments are only annotated for one toxicity label at a time, we ignore other labels than the one annotated in the batch. Furthermore, we made a custom loss function to the model, giving the labels weights in order to tackle the imbalanced label distribution in the data. The weights were calculated based on the labels' frequency in the training data. The resulting weights make the labels with fewer examples in the training data more important to the model and labels with the most examples receive a lower importance. E.g., *threat* received a

weight of 47.6901 due to it appearing in the data only 478 times and *toxicity* the weight 1.4905 due to appearing 15924 times in the data.

No pre-processing for the texts was done to get the best results since previous studies had found that with deep learning pre-processing can make the results worse (Saeed et al., 2018).

For training, we used sequence length of 512 by truncating at the end and did hyperparameter optimization with grid search using learning rate (LR) of (1e-5..5e-5), batch size of (4, 8, 12), and epochs (10) with early stopping and evaluation every 2500 steps. All the hyperparameters were optimized on the development set. For the cross-lingual experiments with XLM-R, we optimized on the English development set and tested on the translated Finnish test set. The best hyperparameters can be found in Appendix B. Furthermore, we used threshold optimization to find the best threshold that maximizes the results for the F1-score.

As metrics in the evaluation, we use micro precision and recall, micro-F1, macro-F1 and ROC_AUC. Precision shows how many of the positive predictions are correct, and recall how many of all the positive cases in the data were found. F1-score is the balanced and harmonic mean of precision and recall. Micro-F1 specifically calculates metrics globally and macro-F1 for each label separately, finding their unweighted mean. Thus, macro-F1 does not take label imbalance into account.

ROC_AUC score is the Area Under the Receiver Operating Characteristic Curve. This metric was used for the scoring of the Kaggle competition held for the original dataset, although only done on the probabilities and 90% of the data as opposed to us using the thresholded label and the full test set.

The codebase for fine-tuning can be found on Github[10] and the fine-tuned model can also be found on Huggingface[11].

## 5 Results

### 5.1 Translation and Transfer

The results of the toxicity detection experiments using the original English and the translated datasets are presented in Table 4. As a baseline, we can consider the results of the English BERT model, 0.69 F1-score (micro-avg.) and 0.89

---

[9]https://huggingface.co/datasets/TurkuNLP/Suomi24-toxicity-annotated

[10]https://github.com/TurkuNLP/toxicity-classifier

[11]https://huggingface.co/TurkuNLP/bert-large-finnish-cased-toxicity

| Model | Train | Test | Precision | Recall | F1-micro | FI-macro | ROC_AUC |
|-------|-------|------|-----------|--------|----------|----------|---------|
| BERT | En | En | 0.59 | 0.81 | 0.69 | 0.61 | 0.89 |
| FinBERT | Fi-DeepL | Fi-DeepL | 0.58 | 0.76 | 0.66 | 0.57 | 0.87 |
| FinBERT | Fi-Opus-MT | Fi-Opus-MT | 0.57 | 0.77 | 0.65 | 0.57 | 0.88 |
| XLM-R | Fi-DeepL | Fi-DeepL | 0.56 | 0.76 | 0.65 | 0.57 | 0.87 |
| XLM-R | En | Fi-DeepL | 0.60 | 0.54 | 0.57 | 0.47 | 0.76 |
| XLM-R | Fi-DeepL+En | Fi-DeepL | 0.56 | 0.78 | 0.65 | 0.57 | 0.88 |
| BERT | Backtr-En | En | 0.59 | 0.77 | 0.67 | 0.60 | 0.87 |

Table 4: Results with different language pairs and models.

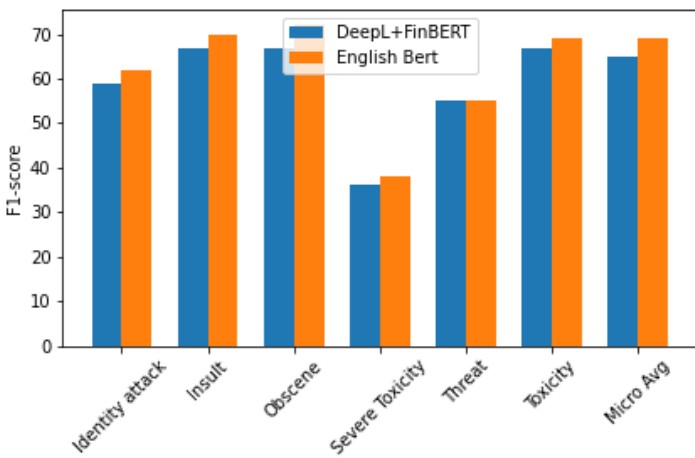

Figure 2: Class-specific F1-scores.

ROC_AUC, trained and tested on the original English data. This is very similar to the results reported by Carta. et al. (2019) using the same Jigsaw dataset and BERT (see Section 2).

FinBERT trained and tested on the machine translated data performs numerically slightly worse than BERT on the English data: 0.66 F1-score with DeepL and 0.65 with OPUS-MT. The loss of performance is, however, very small. With this result, we decide to run the further experiments with the data translated with DeepL.

The multilingual XLM-R performs numerically very similarly to FinBERT with the Finnish DeepL-translated data: 0.65 F1-score. However, its performance is clearly lower when trained on English and only tested on Finnish: 0.57 F1-score. Thus, our results support those by Leite et al. (2020), who noted that zero-shot transfer from English to another language can be challenging.

Our results thus suggest that circumventing the language barrier provides much better results with machine translation than with a cross-lingual model. The quality of the machine translations is further supported by the results on the backtranslated English dataset. By showing only a 2% loss in the F1-score, this experiment supports the qual-

ity of the translations.

Even combining the original English data and its DeepL-translations in the training set does not provide better results than training and testing on the DeepL-translated Finnish data alone, and the model trained and tested in English outperforms both of these settings. This can suggest that transfer, done either with a model or machine translation, can have some effect on the results.

Given the subjectivity associated with toxicity detection, and the IAA scores discussed in Section 2 and our own IAA scores in Section 3.3, the detection results are very close to what can be expected for this task. Additionally, for practical purposes, it is noteworthy that the recall is approximately 20% higher than the precision for all the experiments except for the cross-lingual one. When used for cleaning data or moderating a platform, false positives can be less dangerous than false negatives. This further consolidates the practical usability of the method.

### 5.2 Label-Specific Scores

Nozza (2021) showed that language-specific differences in, e.g., taboo expressions can challenge cross-lingual toxicity detection. These differences

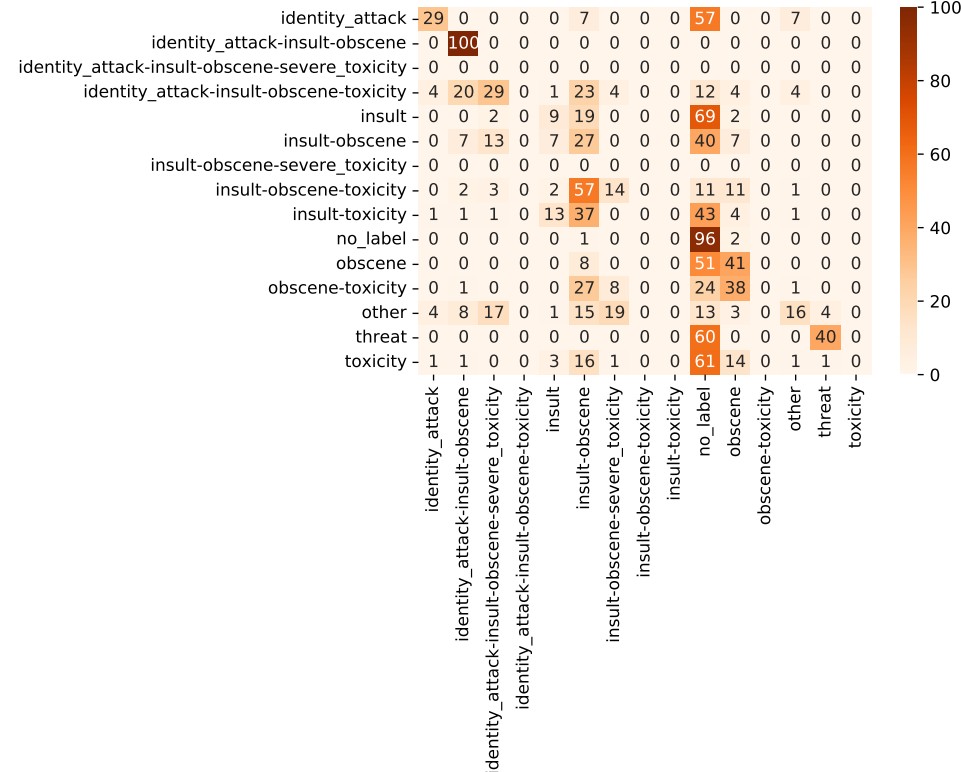

Figure 3: Most frequent classes (rows) and their misclassifications (columns), as percentages of the total number of instances in the data. For the sake of simplicity, co-occurring labels have been fixed as multi-class.

may lead to lower results in particular for some subtypes of toxicity. To ensure that the cross-lingual results we presented in Section 5.1 are not affected by these or similar issues, we inspect label-specific performance metrics. We focus on two models: the best-performing Finnish model trained using FinBERT and the DeepL-translated data, as well as the English model trained using BERT and the original English data.

Figure 2 presents the label-specific metrics obtained using the two models. First, we can see that while the global scores obtained by the English model are slightly higher, the difference remains similar across the labels. Thus, the language transfer does not seem to affect them unevenly.

However, the F1-scores do vary between the different labels. In particular, *severe toxicity* and *threat* receive lower detection scores than the other labels. *Threat* is a very infrequent label, which may also explain its lower detection rate. However, the size of *severe toxicity* is similar to *identity attack*, which nevertheless received better performance. This suggests that the label as such is more vague and less well-defined linguistically.

## 5.3 Error Analysis of the Translated Dataset

Figure 3 presents to which labels the most frequent labels and label combinations were predicted to using the best-performing FinBERT model trained on the DeepL-translated data, and Table 5 presents examples of misclassified comments[12].

The most frequent misclassification for many labels and their combinations is *no label*. In part, this is because of the frequency of comments without any label in the training. On the other hand, this can also reflect the gray zone between different kinds of toxic comments and the ones annotated without a label, prone to subjective interpretation. This can be seen in Example 2 in Table 5, annotated as *insult* and predicted without a label. Whether *hyödytön* 'useless' actually is an insult or not can be prone to subjective interpretations. Furthermore, the Perspective API annotation guidelines in fact note that insult should be targeted towards a person (Perspective, a,b).

Similarly, *severe toxicity* as a label is eas-

---

[12]The examples may be considered offensive. They do not reflect the views of the authors and exclusively serve to explain the linguistic characteristics of the comments.

| ID | Text | True label(s) | Predicted label(s) |
|---|---|---|---|
| FI-1 | *Senkin pikku paskiainen == Senkin pikku paskiainen* | insult, obscene severe toxicity, toxicity | insult, obscene, toxicity |
| EN-1 | You little son of a bitch == You little son of a bitch | | |
| FI-2 | *Wikipedia on täysin hyödytön* | insult | no label |
| EN-2 | Wikipedia is completely useless | | |
| FI-3 | *Painu helvettiin, pedo. Nyljen sinut elävältä, kastelen sinut bensiinillä ja poltan sinut kuoliaaksi, kusipää. Syön vitun sydämesi, senkin pikku mulkku.* | insult, obscene threat, toxicity | insult, obscene severe toxicity, threat toxicity |
| EN-3 | Fuck off paedo. I am going to fucking skin you alive, douse you in petrol then fucking burn you to death, fucktard. I am going to eat your fucking heart you little prick. | | |

Table 5: Examples of misclassified and correctly predicted labels in the DeepL-translated data.

ily not predicted. For instance, *insult-obscene-severe toxicity-toxicity* is frequently misclassified as *insult-obscene-toxicity*. Example 1 illustrates this as well—*severe toxicity* is present in the annotations but not predicted. In this case, the error may be caused by the translation, as the Finnish translation is not as toxic as the original English comment and can even be used to communicate affection. As we mentioned in Section 5.2, *severe toxicity* also received relatively low class-specific scores. Figure 3 shows that it is frequently misclassified as simple *toxicity*. For instance, the label combinations *identity attack-insult-severe toxicity* and *insult-obscene-severe toxicity* are frequently confused with the same labels co-occurring with *toxicity*. Examples 1 and 3 illustrate this as well, as *severe toxicity* is erroneously not predicted for Example 1 and is predicted for Example 3, where it should not have been predicted and the correct label would have been just plain *toxicity* with the other labels.

| | Prec | Rec | F1 |
|---|---|---|---|
| FinBERT-DeepL | 0.57 | 0.59 | 0.58 |
| FinBERT-DeepL Weighted | 0.61 | 0.74 | 0.67 |
| XLMR-En | 0.50 | 0.40 | 0.45 |
| XLMR-En Weighted | 0.50 | 0.40 | 0.45 |

Table 6: Micro evaluation results for the native Finnish dataset using threshold 0.5.

## 5.4 Native Finnish Dataset

We tested the two best-performing models (FinBERT trained on Fi-Depl and XLM-R trained on the original English data) on the native Finnish Suomi24 annotations in order to examine the model performances on texts featuring different language use than the Wiki edit comments included in Jigsaw. The results are presented in Table 6, showing that while the models do find toxic content from the Suomi24 discussions, the performances decrease in comparison with the orig-

inal Jigsaw data (see Section 5.1). Nevertheless, similar to our findings with the Jigsaw data, cross-lingual transfer using a multilingual model provides lower results than a monolingual model trained on translations. Further, similar to the Jigsaw dataset, *severe toxicity* and *threat* received low class-specific scores due to the low amount of examples for those classes in the training data. The metrics for the labels can be found in Appendix C.

A reason for the lower metrics on the Suomi24 discussions can be found in the way the data were sampled (see Section 3.3). By taking even samples from all the prediction score bins even though the large majority of the comments were included in the bins with 0-0.1 or 0.9-1 scores, our sampling method emphasized borderline cases (see Appendix A), and the vast majority of the comments the classifier was certain about were disregarded. The metrics do not take into account this imbalance by default and thus, they can be interpreted rather as macro-average known to display low results for skewed data. Therefore, we counted also the *weighted* metrics using the counts of the bins as weights for the true positive, true negative, false positive and false negative counts. The results achieved using FinBERT-DeepL and this weighing are very similar to those achieved with FinBERT-DeepL on Jigsaw (see Section 5.1).

Table 7 shows examples from the native Finnish dataset. Example 4 presents a comment annotated as *no label*, derived from the bin 0.8-0.9 for *identity attack* predicted by a previous model as a very certain *identity attack* and then later labeled by the new large model as *toxicity*, most likely because the model simply associates 'gay' with toxicity. This illustrates the oversensitivity of the model and bias. A case can be made for the text being intended as an insult but without context that is impossible to say. Example 5 shows a comment binned in very certain *identity attack*, annotated

| ID | Text | Bin | True label(s) | Pred. label(s) |
|---|---|---|---|---|
| FI-4 | *Oletko mahdollisesti homoseksuaali?* | identity attack | no label | toxicity |
| EN-4 | Are you possibly gay? | 0.8-0.9 | | |
| FI-5 | *jos nämä muslimit saavat räjäytellä pommejaan missä haluavat ympäri maailmaa niin miksemme me saa julkaista vitsikkäitä kuvia.* | identity attack | identity attack | identity attack, toxicity |
| EN-5 | if these Muslims can explode their bombs anywhere they want so why can't we publish funny pictures? | 0.9-1 | | |
| FI-5 | *Tästä tulee iso hitti!* | toxicity | no label | no label |
| EN-5 | This is going to be a big hit! | 0.0-0.1 | | |

Table 7: Examples of misclassified and correctly predicted labels in the native Finnish data.

with the same label and then predicted as *identity attack-toxicity*. Here the model succeeds in finding the correct label. Finally, Example 6 presents a comment annotated and predicted as *no label* from the 0-0.1 bin for toxicity—the kind of comment of which the classifier is certain about and our annotation agrees.

## 6 Conclusion

In this paper, we have presented novel resources for Finnish toxicity detection, and we have shown that machine translation is a viable option for circumventing the language barrier for this task. FinBERT and the DeepL-translated data outperformed XLM-R trained on English and tested on Finnish clearly, and the quality of the translation was further confirmed with the backtranslation experiment, showing only a minimal loss in the original English performance. Thus, our results support previous findings by Isbister et al. (2021) and Kobellarz and Silva (2022). Additionally, our results were also confirmed by the results from the native Finnish test set where translation received better results than transfer and our weighted numbers were comparable with the results from using the original translated test set.

The use of machine translation is a cost-effective alternative for building resources when there is no annotated data available in the target language. However, translation can also cause subtle changes in the meaning, which can result in misclassifications and wrong interpretations. Our analysis showed that the toxicity entailed in the original comment can change during the translation to a much less toxic meaning. Therefore, it is crucial that the effect of the translation is evaluated separately for each language and task.

Furthermore, we acknowledge that our model might feature some bias, as illustrated in Section 3.3. Jigsaw has also reported this—the models may learn to incorrectly associate toxicity with, e.g., identities that frequently co-occur with toxic content. This has led to the creation of a new dataset called "Jigsaw Unintended Bias in Toxicity Classification" [13].

In the future, we should further inspect the possible biases the models developed in this study may feature, as well as the model generalizability. Furthermore, multilingual toxicity detection involving code-switching would offer an interesting avenue for the future. Finally, considering the promising results achieved in this study, the use of machine translation for other tasks and language pairs should certainly be analyzed further.

## Acknowledgments

We thank Academy of Finland for financial support and wish to acknowledge CSC – IT Center for Science, Finland for computational resources.

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

## Appendices

## A   The distribution of predicted scores for the Suomi24 data before sampling

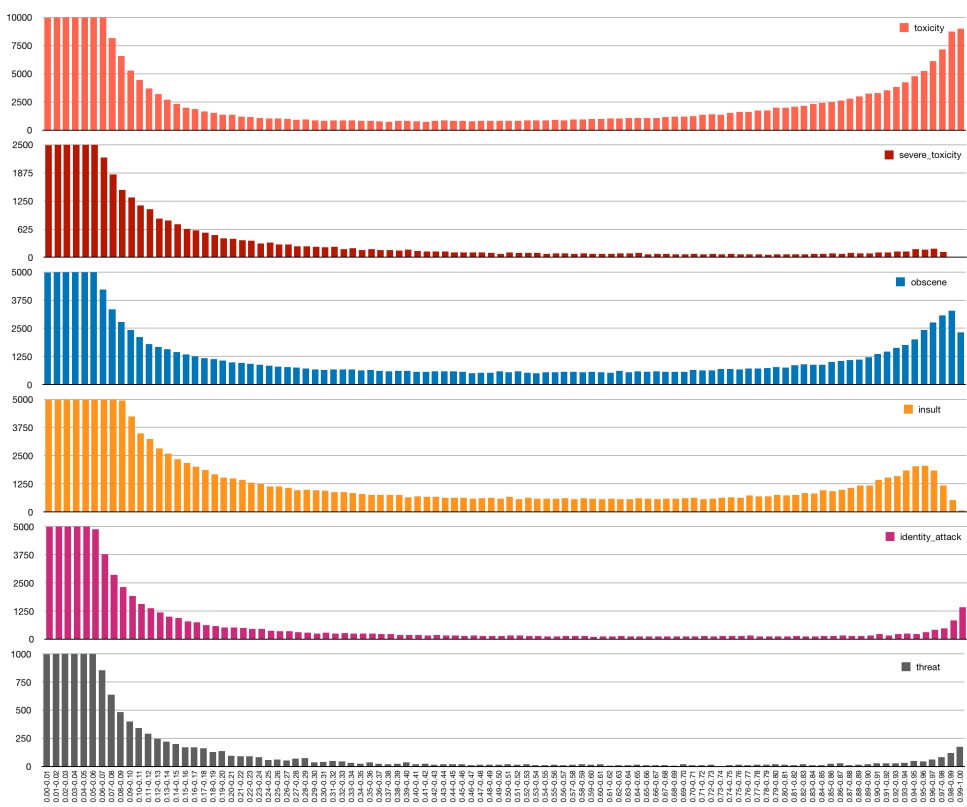

Figure 4: Distribution of prediction scores by label for the Suomi24 data from which our native Finnish dataset examples were sampled for annotation.

## B   Best hyperparamaters for the trained models

| Model | Train | Test | LR | Batch size |
|---|---|---|---|---|
| BERT | En | En | 1e-5 | 12 |
| FinBERT | Fi-DeepL | Fi-DeepL | 2e-5 | 12 |
| FinBERT | Fi-Opus-MT | Fi-Opus-MT | 1e-5 | 12 |
| XLM-R | Fi-DeepL | Fi-DeepL | 1e-5 | 12 |
| XLM-R | En | Fi-DeepL | 1e-5 | 12 |
| XLM-R | Fi-DeepL+En | Fi-DeepL | 1e-5 | 12 |
| BERT | Backtr-En | En | 2e-5 | 12 |

Table 8: Best hyperparameters for each model. Constant parameters were epochs 10 and early stopping 5. Threshold for the labels varied due to threshold optimization during training and evaluation.

## C   Label specific precision, recall and F1 for the native Finnish dataset

| Label | Precision | Recall | F1 |
|---|---|---|---|
| Identity attack | 0.73 | 0.32 | 0.45 |
| Insult | 0.59 | 0.47 | 0.52 |
| Obscene | 0.64 | 0.82 | 0.72 |
| Severe toxicity | 0.12 | 0.29 | 0.17 |
| Threat | 0.32 | 0.29 | 0.30 |
| Toxicity | 0.60 | 0.79 | 0.69 |

Table 9: Micro evaluation results for the labels of the native Finnish dataset using FinBERT-DeepL and a threshold of 0.5.