# OpenReview forum: "Toxicity Detection in Finnish Using Machine Translation"
_NoDaLiDa/2023/Conference — NoDaLiDa 2023_

### Official Review · Reviewer_uymY · 2023-02-27
**Good and very thorough paper on how MT can be used to create language specific resources for toxicity detection.**

**Rating:** 8
**Confidence:** 4

**Review:**

The paper presents a very thorough study of the way MT can be used to circumvent language barriers, specifically to create datasets for toxicity detection in Finnish. The new dataset is created by translating an existing resource for English using two different MT systems, and thoroughly evaluated by running a number of models for toxicity detection on the original resource, the translated versions, as well as  a back-translated English version. In addition, a native Finnish dataset is also compiled, and the best performing model is tested on this new resource.
While the results show that the translated versions provide good datasets on which to train models for toxicity detection in Finnish versions of the original dataset, the change to the new data is not as successful.
Very good evaluation discussion and error analysis.
Some small notes:
l. 85: about using => of how to use
l. 195: Devlin  et al: missing year of publication
l. 228: apply for => apply to
l. 430-433:  Can the change of weights be expressed in a more general fashion?
l. 596 (and other places): transfer the language barrier => circumvent/remove the language barrier



**Paper Type:**

Long paper

---

### Official Review · Reviewer_69RG · 2023-03-13
**Interesting Paper with Value, but Conclusions Are Slightly Unclear**

**Rating:** 7
**Confidence:** 3

**Review:**

The authors investigate the performance of machine translation (MT) of training data ("translate train") for Finnish toxicity detection. They first experiment with a machine-translated test set (using two different MT models), and then look at the performance on a newly collected originally-Finnish dataset.

The paper has a lot of merit, already just because of the new dataset. However, after reading it, I'm a bit unclear about the conclusions. The main experiments show that translating is better than zero-shot crosslingual transfer. However, results on the native Finnish dataset seem to show the opposite, but maybe I'm misunderstanding? Either way is fine, but the message and findings should be clearly communicated.


**Paper Type:**

Long paper

---

### Official Review · Reviewer_f2pc · 2023-03-14
**Interesting study of using translated data for toxicity detection in Finnish, unclear quality of manually annotated testset**

**Rating:** 7
**Confidence:** 2

**Review:**

The paper introduces the first publicly available datasets for toxicity detection in Finnish: A translated version of the English Jigsaw dataset ( setting: multi-label annotation. Its quality is controlled by also using a backtranslated English version) and a smaller set of forum comments originally written in Finnish. The latter one was manually annotated using the same categories as the English Jigsaw (setting: single-label annotation.)

The paper presents experiments on detecting toxicity in Finnish texts:
(a) Translating the training data into Finnish and using the translated data for fine-tuning
(b) using the original English data for fine-tuning and applying zero-shot cross-lingual transfer with XLM-R.
It provides a thorough error analysis and discusses problems of domain transfer (fine-tuning on wikipedia comments vs. testing on general discussion forum comments)

Pros
* First dataset for Finish toxicity detection
* experiments on detection quality of different settings (translated training data vs. zero-shot cross-lingual transfer)
* The paper is well-structured and mostly written in a clear way.

Cons
* There is no information on inter-annotator agreement on the manually annotated data,

Question:
* You followed the label description from the original English guidelines and also created your own guidelines => To what extent did you adapt the guidelines to specific Finnish expressions, structures and cultural issues?
* The manual annotation was done by three annotators: How well did they agree (chance-corrected inter-annotator agreement)?

Minor issue:
* the number formatting is not consistent (e.g., line 112: "223,5492 versus line 350: "945 867")


**Paper Type:**

Long paper

---

### Decision · Program_Chairs · 2023-03-17

Accept